# Mechanisms for the Formation of an Exceptionally Gently Inclined Basal Shear Zone of a Landslide in Glacial Sediments—The Ludoialm Case Study

Xiaoru Dai [1,*], Barbara Schneider-Muntau [1], Julia Krenn [2], Christian Zangerl [3] and Wolfgang Fellin [1]

1 Unit of Geotechnical Engineering, Universität Innsbruck, Technikerstraße 13, 6020 Innsbruck, Austria; barbara.schneider-muntau@uibk.ac.at (B.S.-M.); wolfgang.fellin@uibk.ac.at (W.F.)

2 Department for Road Administration of the Provincial Administration of Lower Austria, Landhausplatz 1, 3109 Sankt Poelten, Austria; julia.krenn@noel.gv.at

3 Institute of Applied Geology, Department of Civil Engineering and Natural Hazards, University of Natural Resources and Life Sciences, Peter Jordan-Strasse 82, 1190 Vienna, Austria; christian.j.zangerl@boku.ac.at

* Correspondence: xiaoru.dai@uibk.ac.at

**Abstract:** The Ludoialm landslide, which is located in the municipality of Münster in Tyrol, Austria, represents a large-scale translational landslide in glacial soil sediments characterised by an exceptionally low inclined basal shear zone of only 12°. Although a temporal coincidence between meteorological events and slope displacement is obvious, the hydromechanical coupled processes responsible for the initial landslide formation and the ongoing movement characteristics have not yet been identified. This article provides a comprehensive analysis of the predisposition factors and the initial failure mechanism of this landslide from geological and geotechnical perspectives. We use a prefailure geometry of the cross section to simulate the initial slope failure process by a limit equilibrium analysis (LEA), a strength-reduction finite element method (SRFEM), and a finite element limit analysis (FELA). The shape and location of the computationally obtained basal sliding zone compare well with the geologically assumed one. Based on the computational study, it turns out that a high groundwater table probably caused by snow melting in combination with different permeabilities for the different layers is needed for the formation of the exceptionally low inclined basal shear zone. This paper presents the failure mechanism of the Ludoialm landslide and discusses the role of the shear band propagation in the process of slope destabilization.

**Keywords:** slope stability; trigger factor; snow melting; seepage; permeability; geotechnical computation

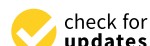



## 1. Introduction

Landslides are a very common natural hazard in mountainous regions and the damage they cause to our lives and property requires more attention [1,2]. Moreover, many landslides develop into cascading effects such as other catastrophic events [3–5]. Due to the variety of factors that trigger landslides and the complex environment in which mountain slopes are often located, the failure and movement mechanisms of landslides are manifold and complex. A deep understanding of landslide mechanisms [6] is essential for the prediction of future landslide events. A key for this is the analysis of slope collapse.

On alpine slopes, deep-seated landslides are a frequently observed phenomenon in Austria, France, Italy or Switzerland [7]. Most deep-seated landslides in the Alps occur on slopes with dip angles around 20–40° [8–13], or at least larger than 15°. The deep-seated basal shear zone in the case study shown here, i.e., the Ludoialm landslide, has an exceptionally gentle dip angle of about 12°, which is even smaller than the soil friction angle of the sliding mass, and is therefore an unusual case compared to other typical landslides and deserves a closer look of its mechanism.

In a mountainous country, snowfall and snow coverage occur regularly in the winter term and lead during snowmelt in spring to a large amount of available water. Slope

instability due to snowmelt is therefore not a rare phenomenon in the world [14–19]. Rainfall or snowmelt events usually increase the pore water pressure in the slope [20–23], and the effective normal stress is subsequently reduced [24,25]. Compared to the usual short duration of rain infiltration, a more continuous supply of water can be provided by the snow-melting process over a longer period [26].

Numerous studies use remote sensing and geographic information systems (GIS) to understand the actual landslide conditions. Field surveys, i.e., geological–hydrogeological mapping, of landslide events and of their historical development process often employ remote sensing methods to derive digital elevation models (DEM), which provide the topography and also help to estimate the landslide geometry, the geological structure of the subsurface, and the volume of the material loss during the events. GIS have widely been used to map and predict landslide hazards and risks in past decades [5,27,28].

The topographical model derived by a field investigation is the basis of the geotechnical model used in the computation. Computational models, which focus on simplified slope geometry and geomechanical processes, can help to understand the slope failure mechanisms. The calculation results can explore some specific aspects of landslide events, e.g., slope deformations, initial failure processes, the volume of the material loss at failure, etc. [29]. The limit equilibrium analysis (LEA) [30,31] is a conventional method for computing the stability of slopes. The advantages of this method are evident, e.g., it is simple and widely applicable. The finite element method [32,33] based on the shear strength reduction concept (SRFEM) [34–36] and finite element limit analysis (FELA) [37] are two state-of-the-art numerical techniques for slope stability analysis [38,39]. In recent years, SRFEM and FELA have been widely used [40–45] to analyse landslides in mountain regions due to changing water tables (e.g., snowmelt, precipitation).

In this work, we focus on the impact of water infiltration due to snowmelt on a slope with a particular hydrogeological situation characterised by two different layers resulting in a large permeability contrast. Few studies have been carried out on the influence of the different permeabilities of soil layers on slope stability [46]. Due to the presence of a relatively high or low permeability layer in some cases [12], several studies have analysed the hydraulic response of soils. Some studies have been conducted on slopes containing relatively permeable layers which are interspersed with or underlain by a low permeability or an impermeable layer [47–49]. The changes of pore pressures or the propagation of excess pore pressures around the interface of layers have been investigated along with the influence on the slope stability. These include some cases in which the upper layer is more permeable than the lower layer, or the case in which the lower layer with a greater permeability is located in a very steep slope. The mechanisms of the effect of the difference in permeability of soil layers on the occurrence of landslide events have not been analysed in detail for these cases. In the case of the Ludoialm landslide investigated in this contribution, the permeability of the lower layer is presumed to be much greater than that of the upper layer based on geological surveys. We show the influence of permeability difference between layers on the Ludoialm landslide in Tyrol, Austria, in order to explain the formation of an exceptionally flat dipping basal shear zone (about 12°). Therefore, we provide a comprehensive analysis of the initial triggering factors of this landslide by analysing the temporal relationship between precipitation, snowmelt, and the events. Photogrammetric methods were used to investigate the historical development of the landslide. A geological and geomorphological mapping was carried out to develop a geometrical–geological model of the slide. A field survey and various laboratory analyses on collected soil samples are performed to obtain the geotechnical properties, i.e., soil cohesion, friction angle, and permeability coefficient. Several slope stability analyses such as LEA, SRFEM, and FELA were applied to identify and validate a plausible initial failure mechanism of the landslide event.

We introduce the study area in Section 2. The identification of the soil parameters is presented in Section 3. We present the numerical studies in Section 4 and the results in

Section 5. Finally, we have a discussion in Section 6 and conclude with the key messages of this study in Section 7.

## 2. Study Area and Data

### 2.1. The Ludoialm Landslide

The Ludoialm is located in the municipality of Münster in Tyrol, Austria (Figure 1), where the Northern Calcareous Alps form the regional geological framing. The landslide area is mainly pasture and partly covered by forest. Except for the periods of reactivations with high velocities, the landslide could be classified as a very slow to slow planar soil slide [50]. According to geological surveys in 2014 and before, the landslide was moving at a certain speed every year, e.g., approximately 0.6 m per year between 1992 and 2004 [51].

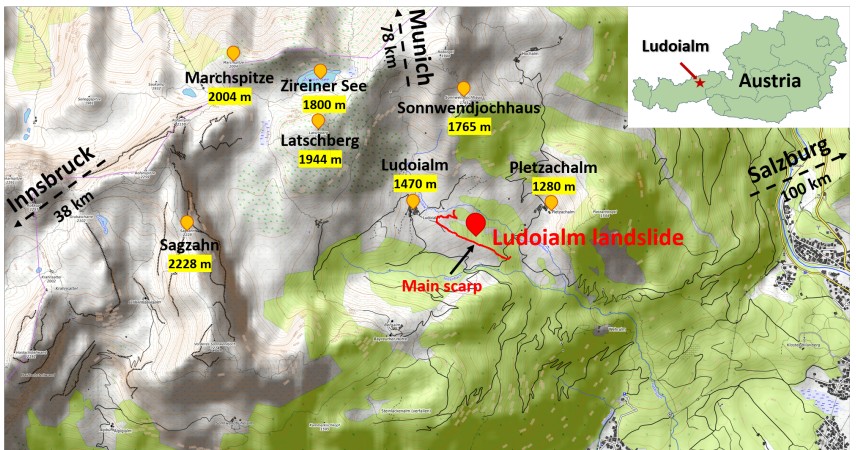

**Figure 1.** The location and surroundings of the Ludoialm landslide (Latitude: $47°27'10.86''$ N; Longitude: $11°49'47.12''$ E) (OpenTopoMap).

The initiation of the landslide is dated to 1952 according to aerial views, historic information, and maps [51]. Two remarkable reactivations of the Ludoialm landslide occurred on 15 April 1967 and 5 February 1999, in the past decades. The landslide event in 1967 was most probably due to intensive snow melting. Furthermore, several, either active or inactive, rotational secondary slides have occurred at the toe of the landslide area, leading partly to debris flow events.

The horizontal length of the landslide is about 550 m, the maximum width is 180 m, and the maximum thickness is about 20 m. After the two reactivation events, a material loss of approximately 486,000 $m^3$ in the landslide area can be estimated by the analysis of GIS data. The basal sliding surface is roughly parallel to the slope surface, with a small dip angle of only about 12° based on the geological survey.

### 2.2. Geological and Hydrogeological Setting

Due to the complex structure of nappes and the large variety of different sedimentary rocks of the Northern Calcareous Alps, only the lithological units, directly influencing the formation of the Ludoialm slide, were examined. In the landslide area, Cretaceous sediments of the Gosau Group were mapped, characterised by flat bedding planes. The sequence of the Cretaceous layers at the toe of the landslides has already been described in detail by Ampferer [52], showing a stratification from the bottom to the top based on a layer of (1) coal, (2) blue-grey sandstone, (3) gray marl with gastropods, (4) sand with corals and gastropods, (5) grey marl with gastropods, (6) breccia with components of rudist limestone, (7) grey clay with components of white shells, (8) solid limestone with sandstone, (9) grey clay, partly petrified, and (10) lime- and sandstone. A layer of glacial deposits with gravel to boulders showing scratch marks, e.g., lodgement till, were deposited on an unconformity formed by erosional processes on top of the Gosau sediments. The bulk mineralogical composition of the uppermost Gosau sediments, i.e., a clay-rich layer which

represents the boundary with the glacial deposits, was determined by X-ray diffractometer (XRD) measurements. The high content of calcite, dolomite, and sheet-silicates indicated a marly composition, with illite, chlorite, kaolinite, and vermiculite as main clay minerals.

In general, the uppermost layer of the Gosau sediments and its contact with the glacial deposits is essential for an understanding of landslide processes, since the extraordinary flat dipping basal shear zone is located within this zone. Decomposed, weathered, and heavily fractured marls were mapped at least where the contact is outcropping at the toe of the slide. The marls showed a wide range of states depending on the degree of weathering and water content, characterised by a range of moderately weathered rock to a soft and clay-rich residual soil with high plasticity. Figure 2 shows different weathering grades of Cretaceous marl by the field investigation. Given that water is present in the entire landslide area, these residual soils are mainly encountered in the uppermost layer which was later on overlaid by the glacial deposits.

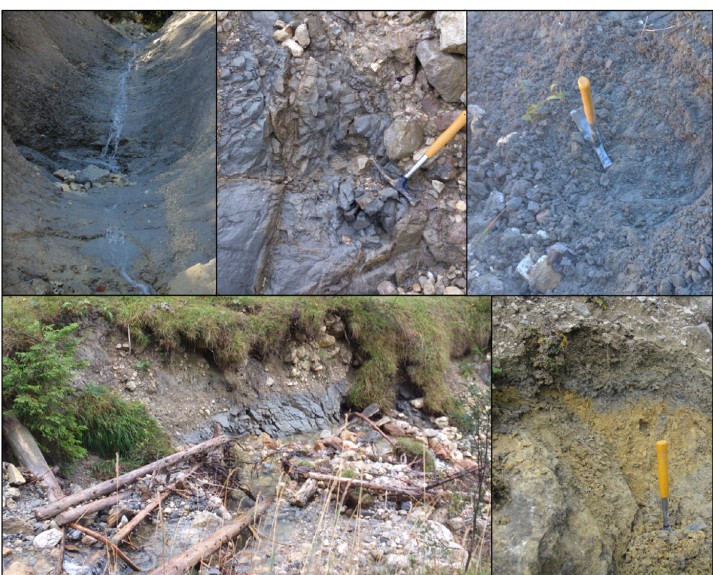

**Figure 2.** Different weathering grades of Cretaceous marl at the Ludoialm landslide area by field investigation (from Krenn [51]).

Sandstone layers interstratified within the marls ranging from decimetres to metres in thickness were found during the geological field survey within and outside of the landslide area. Due to the lack of drillings, the spatial extent of the sandstone layers is unclear but significant to understand the hydrogeological situation, for controlling the landslide failure, and the deformation behaviour. It is assumed that the interstratifications of sandstone provide permeable zones within the clay-rich Gosau sediments and thus affect the pore pressure distribution at the basal shear zone.

Glacial deposits, which essentially form the landslide mass, represent the product of local glaciation characterised by low transport distances. Therefore, evidence was found in the degree of rounding and shape from gravel to boulder as well as the occurrence of rocks that could be stratographically assigned to the surrounding area. The fine-grained matrix of the glacial deposits consists of silty-clayey materials with a varying content of sand. Figure 3 shows different outcrops of glacial deposits at the landslide area. Stratification within the glacial deposits was locally observed in the scarp area, suggesting that debris flow activity may have caused to some extent the relocation and deposition of talus and reworked glacial deposits.

Numerous ponds and waterlogging zones were mapped on the landslide surface, indicating a low permeability of the area. A surface drainage system based on trenches and pipes was installed after the event of 1967 in order to reduce the impact of infiltrating water on slope movement. Remarkably, springs were not mapped on slopes surrounding the landslide and scarp faces. The lack of such hydrogeological indicators and piezometer

installations makes it difficult to create a profound hydrogeological conceptual model of the landslide site. However, in situ and laboratory testing in combination with scenario studies based on hydromechanically coupled numerical modelling provide valuable insight into possible failure and deformation mechanisms.

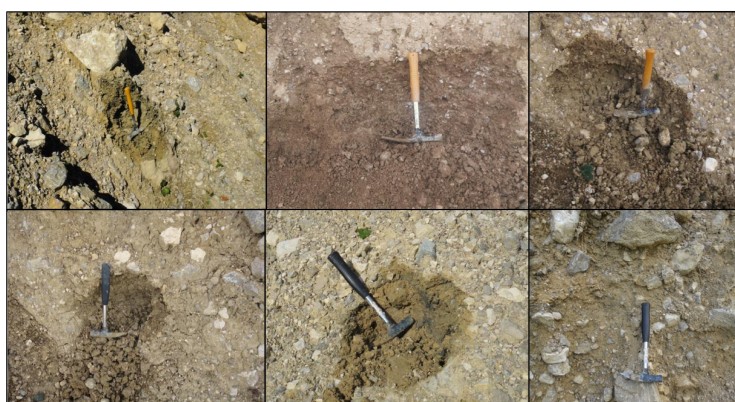

**Figure 3.** Different outcrops of glacial deposits at the Ludoialm landslide area (from Krenn [51]).

*2.3. Photogrammetric Analyses*

The preprocessing steps were carried out on the raw data from aerial photographs of the project area, which were ordered from the Federal Office for Metrology and Surveying (BEV) and the map service of Tyrol (TIRIS). The photogrammetric processing of image pairs was carried out in ERDAS Imagine 2014 to produce digital surface/elevation models (DSM/DEM) and orthophotos for different years. Airborne laser-scanning (ALS) digital elevation model and orthophotos are available for the years 1965, 1973, 1974, 1992, 1997, and 2004/2007.

The volume of total material loss removed from the landslide area by debris flow activity was estimated by the difference between the prefailure topography and the postfailure topography. There was no available DEM information for the prefailure topography. Therefore, a DEM of the prefailure topography was established by reshaping the contour lines which were obtained by the ALS DEM of 2007. The cut/fill function (spatial analyst) in ArcGIS 10 was used to obtain the comparison between the obtained DEM for the prefailure topography and the 2007 DEM for the postfailure topography. The reshaping process was conducted in a conservative way to avoid an overestimation of the landslide volume. Only the clearly recognizable material loss was equilibrated. Figures 4 and 5 show the analyses of material loss for different zones in the landslide area.

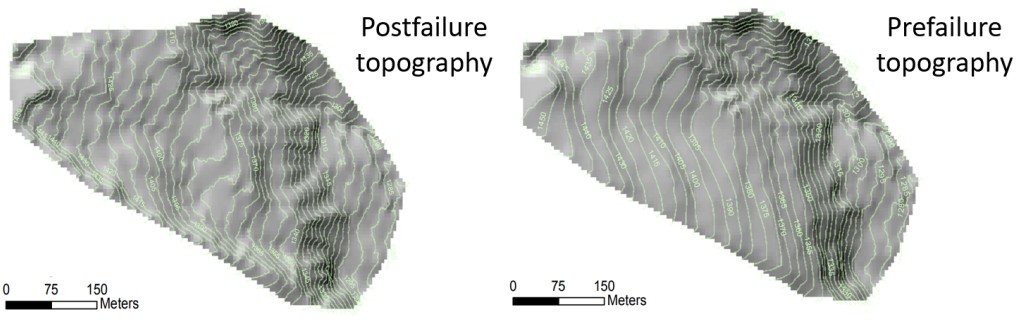

**Figure 4.** Analysis of material loss by DEM models including postfailure topography (2007 DEM) and established prefailure topography (modified from Krenn [51]).

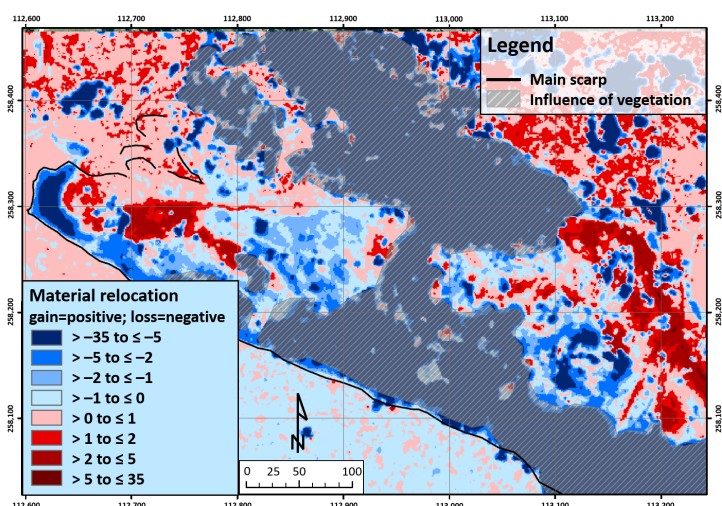

**Figure 5.** Material gain/loss in the landslide area in the period between 1965 and 2007 (modified from Krenn [51]).

### 2.4. The Cross Section

A representative cross section near the centreline along the landslide direction was selected for the computations (Figure 6). The digital elevation model (DEM) of the prefailure topography (Figure 7a) [53] was constructed from the postfailure geometry (Figure 7b) [51] by using all geological, geometric, and kinematic findings of the landslide, e.g., geological investigations, photogrammetric analysis, and various analyses of raster and point data from GIS.

The uppermost layer consists mainly of glacial sediments, which deposited on marls of the Gosau group. The nomenclature "glacial sediments" is not entirely correct in the Ludoialm landslide area as rock fall and multiphase debris flow processes may have modified the glacial sediments. Glacial till was mixed, at least locally, with angular debris flow deposits either during one single event or throughout multiple events. The thickness of the resulting deposits varied from tens of metres to a few decimetres. The assumed basal shear zone lay partly between both layers and partly in the marl with an inclination of approximately 12° according to the geological survey. Our simulation model was based on the reconstructed prefailure geometry.

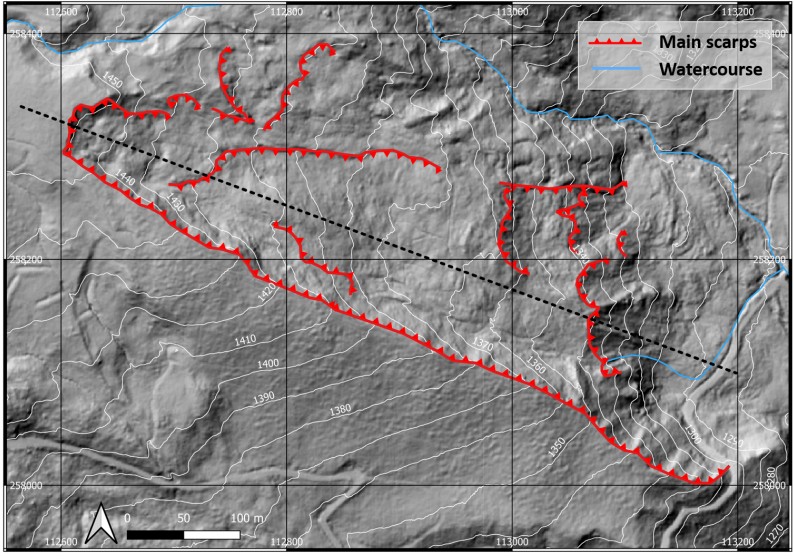

**Figure 6.** Location of the cross section (black dotted line).

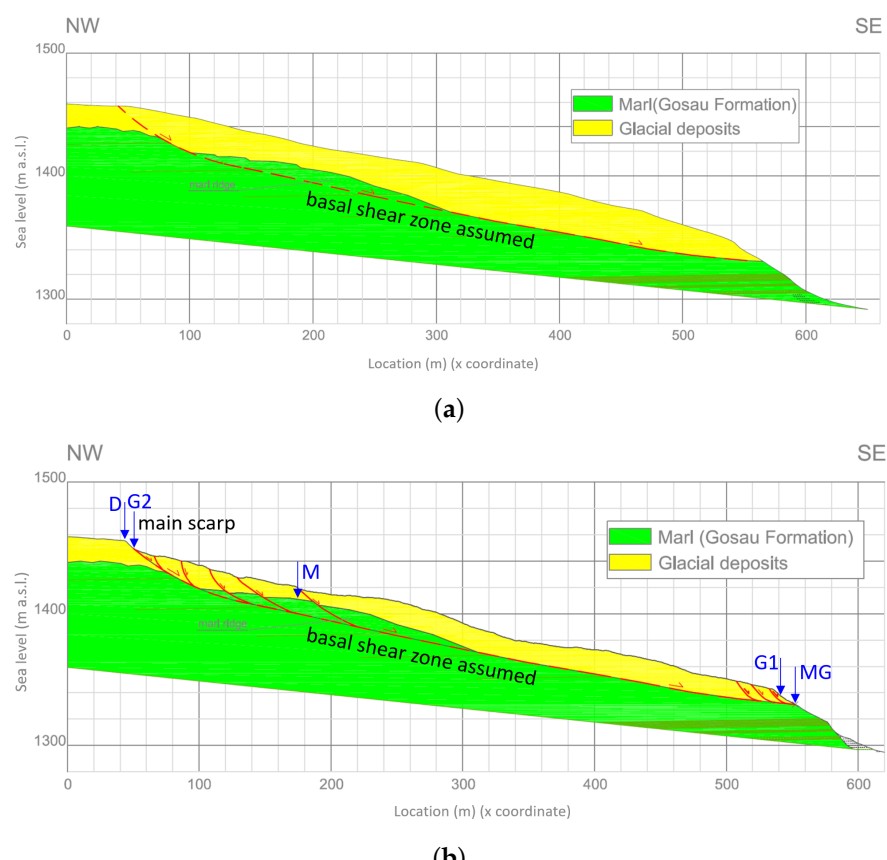

**Figure 7.** Simplified models of the slope (modified from Krenn [51] and Renner [53]). (**a**) Prefailure geometry of the Ludoialm landslide; (**b**) postfailure geometry of the Ludoialm landslide and locations of soil samples projected into the cross section of Figure 6. Note that the samples were not taken directly in the line of the cross section but laterally from it.

## 3. Parameter Identification

This section provides a detailed explanation of the determination of the most important parameters, i.e., strength parameters and permeability coefficients. In Schneider-Muntau et al. [54], the influence of the input parameters was elaborated as a sensitivity analysis.

### 3.1. Strength

The soil samples were taken from different locations of the landslide area. Figure 7 shows the locations of soil samples D and G2 around the main scarp and soil samples G1 and MG at the toe projected into the cross section. Note that the samples were not taken directly in the line of the cross section but laterally from it [51], which took into account the impact of heterogeneous characteristics of the glacial deposits on the mechanical behaviour. Consolidated drained triaxial tests (CD) were conducted and the resulting Mohr–Coulomb parameters are shown in Table 1. Soil samples G1 and G2 were glacial deposits from the uppermost layer. Sample D was also from the upper layer; however, the material was debris flow deposit, which represented only a small percentage of the soil in the upper layer. Sample M was marl from the Gosau Group exposed at the right later flank of the landslide [51]. Soil sample MG was taken at the boundary of glacial deposits and marl, representing a mixture of both. In the consolidated drained triaxial tests, the lateral pressure ($\sigma_3$) was kept at 300 kPa, 600 kPa, and 900 kPa; see Figure 8.

**Table 1.** Resulting Mohr–Coulomb parameters of consolidated drained triaxial tests by [51].

| Sample Name | Material Type | $c'$ (kN/m$^2$) | $\varphi'$ (°) |
|:---:|:---:|:---:|:---:|
| G1 | Glacial deposits | 39 | 16.8 |
| G2 | Glacial deposits | 0 | 18.8 |
| D | Debris flow deposits | 5 | 40.0 |
| MG | Marl–glacial sediments (boundary layer) | 52 | 21.5 |
| M | Marl | 96 | 27.6 |

It can be concluded from Table 1 that marl had a higher cohesion than glacial deposit. The strengths of the glacial deposits taken from different locations varied widely. The particle size distribution tests were performed at the laboratories of the University of Natural Resources and Life Sciences in Vienna (BOKU) and the University of Innsbruck (UIBK). The particle size was classified according to ÖNORM L 1061-1 [55] and ÖNORM L 1061-2 [56], and the percentages of each grain size class (clay, silt, sand, and gravel) are shown in Table 2. The content of clay and silt for the glacial deposits usually exceeded 50%. The fraction of clay plus silt of sample D, i.e., debris flow deposits, was very low, about half as much as in the glacial deposit samples, while the content of gravel was very high, which well explains the high friction angle of this sample. To summarize all these characteristics, the soil sample D embodied very different characteristics from other ones also taken from the uppermost layer and constituted only a small portion of the upper layer. This sample was therefore not taken into account in the computation.

**Table 2.** Resulting classification of materials from particle size analysis.

| Sample | BOKU (Unit: %) | | | | | UIBK (Unit: %) | | | | |
|:---:|:---:|:---:|:---:|:---:|:---:|:---:|:---:|:---:|:---:|:---:|
| | Clay | Silt | Sand | Gravel | Clay + Silt | Clay | Silt | Sand | Gravel | Clay + Silt |
| G1 | 28.3 | 33.4 | 13.8 | 24.5 | 61.7 | 21.8 | 28.6 | 20.9 | 28.7 | 50.4 |
| G2 | 29 | 37.7 | 21.8 | 11.5 | 66.7 | 24.7 | 34.3 | 20.5 | 20.5 | 59 |
| D | 9.9 | 22.4 | 26.7 | 41 | 32.3 | 5.8 | 20.4 | 27.1 | 46.7 | 26.2 |
| MG | 33.7 | 44.6 | 12.7 | 9 | 78.3 | 24.9 | 35.4 | 17.1 | 22.6 | 60.3 |
| M | 22.8 | 64 | 13.2 | 0 | 86.8 | 22.4 | 60.9 | 16.4 | 0.3 | 83.3 |

Since sample MG was taken from the boundary of glacial deposit and marl, its cohesion was expected to be higher than the samples of pure glacial deposits. However, it was heavily weathered and its strength was closer to that of the upper layer, so it was taken into account in the determination of the strength for the upper layer. The cohesion of G2 was remarkably lower than expected for glacial deposits, which may be due to the fact that it was taken near the main scarp of the slope. We took a computational mixture of the soil samples to represent the strength parameters of the upper layer, i.e., the sliding mass. The parameter combination of MG, G1, and G2 was used for the upper layer. Based on the linear fitting method and under consideration of all stress levels [57,58], the peak shear strength parameters of the Mohr–Coulomb failure criterion were computed from the results of triaxial experiments, as listed in Table 3 and shown in Figure 8. As for the marl layer, only one sample was available, and its values were used in the computation.

**Table 3.** The geotechnical parameters of the Mohr–Coulomb criterion for both layers (mean values for upper layer).

| Soil Layer | Sample | $c'$ (kN/m$^2$) | $\varphi'$ (°) |
|:---:|:---:|:---:|:---:|
| Upper | MG, G1, and G2 | 31.4 | 19.1 |
| Lower | M | 96 | 27.6 |

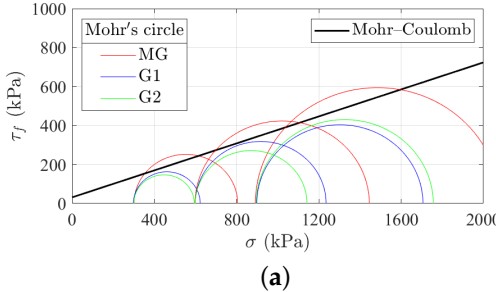
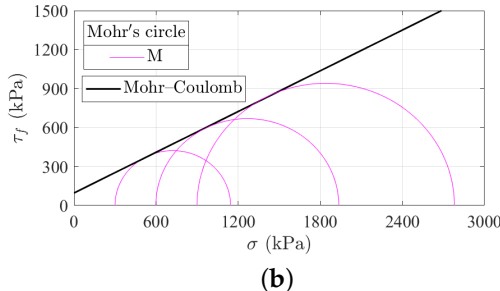

**(a)**            **(b)**

**Figure 8.** Mohr's circles of soil samples and resulting shear strengths of Mohr–Coulomb. (**a**) Samples MG, G1, and G2 (in total 9 stress levels) for the upper glacial deposit layer; (**b**) sample M (in total 3 stress levels) for the lower marl layer.

Given the scatter of experimental results, the upper and lower bounds of a 90% confidence interval of the regression were used to estimate realistic bounds for the shear parameters of the upper layer. In engineering, characteristic shear parameters, i.e., conservative estimates of the mean values, are used, which in a statistical sense represent a 5% fractile of the strength. Figure 9 shows the 90% confidence region obtained by a Bayesian method [57–59] and the resulting linearization in the stress range of the experiments. The Bayes method allows one to include empirical limits for the strength parameters. We used a relatively large range as limits here in order not to influence the estimation too much: from 0 kPa to 100 kPa for the cohesion and from 0° to 45° for the friction angle, which can be considered as extreme limits for glacial deposits. The blue solid lines in the figure correspond to the 5% and 95% fractiles. These were computed by linearizing the boundaries of the confidence interval. Table 4 shows the shear parameters corresponding to the 5% fractile and 95% fractile shear strength for the upper layer.

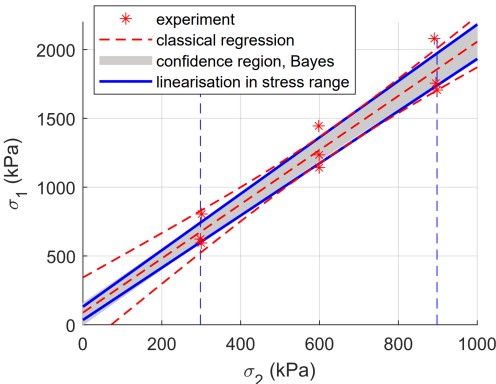

**Figure 9.** Result of the Bayesian approach based on experimental data of samples MG, G1, and G2. The stress range used for linearization is depicted by vertical blue dashed lines. $\sigma_2$ is the lateral stress given by the confining pressure in the consolidated drained triaxial test, and $\sigma_1$ is the axial stress by measurement in the CD test.

**Table 4.** Limits for the shear parameters as 5% and 95% fractiles of the shear strength of the upper layer.

|  | $c'$ (kN/m$^2$) | $\varphi'$ (°) |
|---|---|---|
| 5% fractile | 12.5 | 18.1 |
| 95% fractile | 45.8 | 20.2 |

### 3.2. Permeability

Laboratory tests and field tests were carried out on the samples, which were the falling head permeability test and ring infiltrometer test, respectively. For a ring infiltrometer test, one ring is rammed into the ground and is filled with water. Water is added until

reaching a quasi-stationary flow to keep the water level steady. The hydraulic conductivity is backcalculated by monitoring the time and the amount of added water [60]. The resulting coefficients of permeability for different layers are listed in Table 5.

**Table 5.** Hydraulic conductivity rates measured by the tests (modified from [51,53]).

| Soil Layer | Sample | Coefficient of Permeability (m/s) |
|:---:|:---:|:---:|
| Upper | MG | $9.1 \times 10^{-11}$ (lab test) |
| Lower | M | $5.11 \times 10^{-9}$ (field test) |

For the upper layer, the soil experienced long travel distances in its emergence. Therefore, no predominant discontinuities were expected or observed during the field campaign. Reconstituted samples on homogeneous material should reveal realistic values for the permeability. There are also many empirical relations available for determining the permeability coefficient based on the particle size data, e.g., Kozeny–Carman formula, Breyer formula, Terzaghi formula, etc. [61–70] which can be used for validation. These empirical formulas have limitations. The $d_{10}$ (10% of the weight of the soil sample is finer than this grain diameter) of our samples in the upper layer was very small and the uniformity coefficient was relatively large, so the conditions for most empirical relations were not fulfilled. However, the Kozeny–Carman (KC) equation for estimating the soil permeability was appropriate for this case and utilized for the validation to ensure the plausibility of the experimental data:

$$k = \frac{g}{\nu} \times 8.3 \times 10^{-3} \left[ \frac{n^3}{(1-n)^2} \right] d_{10}^2 \tag{1}$$

Therein, $k$ is the permeability; g is the acceleration due to gravity; $\nu$ is the kinematic viscosity, which can be obtained from the dynamic viscosity ($\mu$) and fluid (water) density ($\rho$):

$$\nu = \frac{\mu}{\rho} \tag{2}$$

The porosity $n$ can be derived from the empirical relationship [63]:

$$n = 0.255(1 + 0.83^{C_u}) \tag{3}$$

The coefficient of uniformity $C_u$ is given by: $C_u = d_{60}/d_{10}$ ($d_{60}$ and $d_{10}$ are the grain diameters for which 60% and 10% (by weight) of the sample is finer, respectively).

According to the data from the particle size distribution test, the permeability coefficients of different samples in the upper layer could be derived by the Kozeny-Carman equation with the temperature of 0° or 10°, for example, as shown in Table 6.

**Table 6.** Permeability coefficients of soil samples by the Kozeny–Carman equation for the upper layer.

| Temperature (°C) | Coefficient of Permeability (m/s) | | |
|:---:|:---:|:---:|:---:|
| | **MG** | **G1** | **G2** |
| 0 | $7.6 \times 10^{-11}$ | $5.4 \times 10^{-11}$ | $7.2 \times 10^{-11}$ |
| 10 | $1.0 \times 10^{-10}$ | $7.4 \times 10^{-11}$ | $9.8 \times 10^{-11}$ |

The permeability coefficients of the samples MG, G1, and G2 calculated by the Kozeny–Carman equation were of the same order of magnitude and very close to each other (Table 6). These values were also close to the experimentally obtained permeability coefficients (Table 5), so it was considered reasonable to use $9 \times 10^{-11}$ m/s as the coefficient of permeability for the upper layer in the calculation.

Fractures, fissures, tension cracks, and other potential channels existed in the lower marl layer. Water seepage took place in the field test, most probably due to infiltration

along the discontinuities in the marl. The field test could be considered reasonable, and we used $k = 5 \times 10^{-9}$ m/s for our computations.

## 4. Numerical Studies

Three methods commonly used in slope stability analysis were employed and compared: the limit equilibrium analysis (LEA), strength-reduction finite element analysis (SRFEA), and finite element limit analysis (FELA).

**Limit equilibrium analysis:** The LEA is widely used due to its simplicity and long history. The slope stability analysis is conducted with curved slip lines (e.g., Bishop, Janbu, Spencer) to get the factor of safety (FoS). This method has been implemented in some commercial software, such as Rocscience [71]. Although this method does not satisfy the overall equilibrium conditions, e.g., several assumptions regarding the interslice forces need to be made in advance, it is easy to handle and has been validated in case studies [5,54].

**Strength reduction finite element analysis:** The finite element code Plaxis [72] was used for the slope stability analysis of the case study in this application. The FoS was obtained by the strength-reduction method (SRM) [32]. In this approach, the shear strength parameters ($c$ and $\varphi$, if a Mohr–Coulomb failure criterion is assumed) were reduced until the slope failure occurred [73]. The definition of the FoS is:

$$\text{FoS} = \frac{c}{c_{\text{red}}} = \frac{\tan \varphi}{\tan \varphi_{red}} \tag{4}$$

where $c$ and $\varphi$ are the actual material strength parameters, and $c_{\text{red}}$ and $\varphi_{\text{red}}$ are the reduced parameters for which no equilibrium can be found any more.

**Finite element limit analysis:** A FELA is based on the theorems of plasticity developed by Drucker et al. [74]. In the past decades, the finite element formulations of the upper- and lower-bound theorems of plasticity were significantly developed and successfully applied to slope stability problems [37–39]. The method can calculate the upper and lower bound of the FoS based on the formulations developed by Sloan [37]. For instance in the software application OptumG2 [75], a common definition for the FoS involves the shear strength of the material related to the strength that will cause the slope collapse, which is analogous to the strength-reduction method elaborated in the finite element method (as illustrated in the above paragraph).

### 4.1. Boundary Conditions

As mentioned in Section 2.4, the calculation model was established based on the prefailure geometry. The cross section in Figure 7a shows an altitude approaching 1460 m a.s.l. at the left boundary. The slope in the calculation was modelled from roughly 1300 m a.s.l. to the surface of the slope. The thickness of the upper glacial deposits was between 20 and 30 m.

The left and right boundaries were set far away from the occurred sliding mass. Each soil layer of the model was considered to be homogeneous in the calculation scale, and the horizontal and vertical permeability coefficients were set as equal values.

### 4.2. Material Properties

Table 7 summarizes an overview of all input parameters used in the calculation. The moduli of elasticity were taken from the results of laboratory test [51] and a literature review. The modulus used for the upper glacial deposit layer was the average secant Young's modulus (around 20 MPa) evaluated at 50% of the compressive strength [76] by triaxial tests. The elastic modulus of the lower marl layer was larger than 30 MPa on the reconstituted and disturbed samples in the laboratory triaxial test. However, these few experimental data may not represent the whole area of the marl layer, and we used 100 MPa here, based on the literature review on weathered marl [77–80]. The values of Poisson's ratio used in the computation were also determined from the volumetric behaviour in the tests at 50% of the compressive strength with the following relationship:

$$\nu = -\frac{\varepsilon_3}{\varepsilon_1} = \frac{1}{2} - \frac{\varepsilon_v}{2\varepsilon_1} \tag{5}$$

Therein, $\varepsilon_1$ and $\varepsilon_v$ represent axial and volumetric strains at 50% of the compressive strength, respectively. It was verified by computation that Young's modulus or Poisson's ratio had almost no effect on the results in this case.

The analyses considered drained conditions and applied a linear–elastic perfectly plastic constitutive model with a Mohr–Coulomb failure criterion. A nonassociated flow rule ($\psi = 0$) was adopted. The reduced shear parameters in the LEA were obtained by the following equations [81] to account for the nonassociated flow rule, which were introduced by Davis [82]:

$$c^* = c\frac{\cos\varphi\cos\psi}{1 - \sin\varphi\sin\psi} \tag{6}$$

$$\tan\varphi^* = \frac{\sin\varphi\cos\psi}{1 - \sin\varphi\sin\psi} \tag{7}$$

**Table 7.** Material parameters for both soil layers.

|  | Unit | Upper Layer | Lower Layer |
|---|---|---|---|
| $\gamma$ | (kN/m$^3$) | 20 | 23 |
| $E'$ | (MPa) | 20 | 100 |
| $\nu'$ | (-) | 0.4 | 0.4 |
| $c'$ | (kPa) | 31.4 | 96 |
| $\varphi'$ | (°) | 19.1 | 27.6 |
| $\psi$ | (°) | 0 | 0 |
| $c^*$ | (kPa) | 29.7 | 85.1 |
| $\varphi^*$ | (°) | 18.1 | 24.9 |
| $k$ | (m/s) | $9 \times 10^{-11}$ | $5 \times 10^{-9}$ |

*4.3. Hydraulic Conditions*

Precipitation and snow melt affect the groundwater level; therefore, analyses without and with seepage were carried out. As the snowmelt takes time and the corresponding rise in the groundwater level does not change dramatically in a short period, a steady-state analysis for the seepage condition was adopted. The height of the groundwater level was unavailable when the landslide occurred, so different groundwater tables were chosen in the calculations to see the failure mechanism for different cases. The location of the groundwater table at the leftmost boundary was defined as the initial groundwater table in this contribution.

Three initial groundwater tables were applied in the calculation, which were 1400 m (GWT1), 1430 m (GWT2), and 1458 m (GWT3), respectively (Table 8). Compared to GWT1, the average height of GWT2 was increased by roughly 21 m, and the average height of GWT3 was increased by about 36 m. The largest part of the assumed basal shear zone was above the groundwater table for the lowest one (GWT1), and it was below the groundwater table for the middle one (GWT2). The highest groundwater table (GWT3) was almost approaching the slope surface. The stability analyses for the no-seepage condition and seepage condition from the lowest to the highest level could basically reflect the effects of seepage and changes in pore water pressure on the slope stability and the range of instability due to snow melting and possible precipitation. The changes in the pore pressure distribution at the location of the basal shear zone is explored in Section 5.

**Table 8.** FoS values obtained with different groundwater levels (LB: lower bound; UB: upper bound).

| Initial Groundwater Table | LEA | SRFEA | FELA | | |
| --- | --- | --- | --- | --- | --- |
| | | | **LB** | **UB** | **(LB + UB)/2** |
| 0 | 1.30 | 1.37 | 1.37 | 1.39 | 1.38 |
| GWT1-1400 m | 1.19 | 1.27 | 1.33 | 1.35 | 1.34 |
| GWT2-1430 m | 1.10 | 1.16 | 1.23 | 1.24 | 1.24 |
| GWT3-1458 m | 0.93 | 0.94 | 1.00 | 1.01 | 1.01 |

## 5. Results

Vertical slices were used in the LEA, and the results were obtained with the Spencer method. In the limit equilibrium model for the steady-state groundwater analysis, the approximate number of elements was 6000, and the element type was chosen as three-node triangles. In the finite element model, the number of elements approached 6000 and the mesh discretization was performed with 15-node triangular elements. Cases with FoS < 1 were computed with doubled starting strength parameters. The final safety factor was obtained by dividing the resulting value by a factor of two. In the finite element limit analysis, the strength-reduction method was used for the lower- and upper-bound calculations. In the computational model for the finite element limit analysis, the number of start elements was set as 6000 with mesh adaptivity. Very dense meshes were adopted for the area surrounding the assumed basal shear zone in the computational models with the strength-reduction finite element analysis and finite element limit analysis. It was computationally verified by a convergence study that the grid density settings in the models used in the three methods could obtain comparatively reliable results without excessive consumption of computational time.

Figure 10 shows the mesh discretization and failure mechanisms without considering seepage obtained by the three methods, in which (a), (b), (c), and (d) correspond to the results of the limit equilibrium analysis, strength-reduction finite element analysis, and the lower and upper bound of the finite element limit analysis, respectively. It can be seen that the location of the shear zone occurred at the toe of the slope when seepage was not considered (FoS > 1).

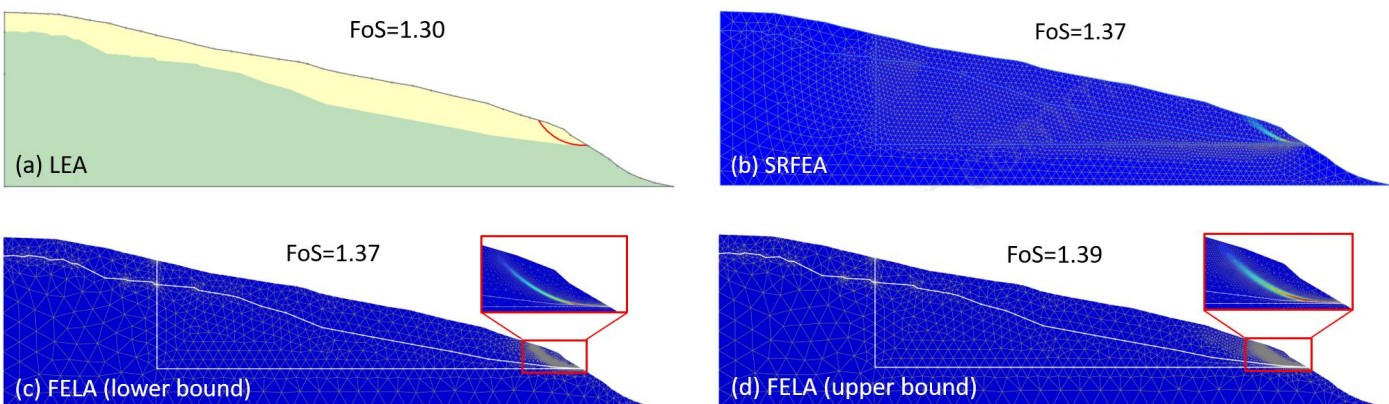

**Figure 10.** Safety analysis without the consideration of seepage: (**a**) sliding surface obtained with the LEA; (**b**) incremental shear strains obtained with the SRFEA; (**c**) shear dissipation distribution obtained with FELA (lower bound); (**d**) shear dissipation distribution obtained with the FELA (upper bound).

The calculation results for the groundwater table at its highest position are provided here as an example in Figure 11. At the left boundary of the computational model, the groundwater table was set at a fixed height that did not exceed the leftmost altitude, which was 1458 m. Figure 11 indicates that the slope was in an unstable state with its safety factor close to or less than one for this high groundwater level. The sliding surface appeared

mainly at the boundary of both layers in the calculation. The sliding area was quite large, starting from the upper middle of the slope, almost through the entire upper soil layer.

The safety factors obtained by the three methods under no-seepage and seepage conditions are listed in Table 8. The FoS values indicated that the safety factor gradually decreased with the rise in groundwater levels until the slope became unstable. Figures 10 and 11 and Table 8 show how similar the FoS values and the failure mechanisms obtained with the three methods were.

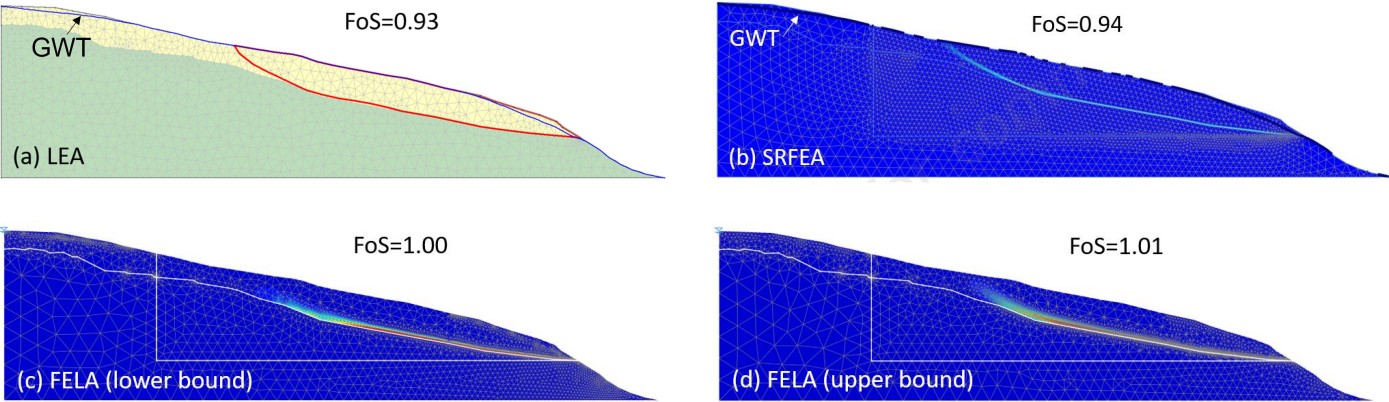

**Figure 11.** Safety analysis with the initial groundwater table at 1458 m: (**a**) sliding surface obtained with the LEA; (**b**) incremental shear strains obtained with the SRFEA; (**c**) shear dissipation distribution obtained with FELA (lower bound); (**d**) shear dissipation distribution obtained with the FELA (upper bound).

The calculation results obtained with the limits of the Mohr–Coulomb shear parameters as a 5% fractile and a 95% fractile under no-seepage and seepage (the initial groundwater table at GWT3-1458 m) conditions, are listed in Table 9 and shown in Figure 12. Only the results of the calculated sliding surface by the limit equilibrium analysis are shown here since the locations obtained by the two other methods were similar. The results demonstrated that the different shear parameters affected the size of the sliding body and much more the calculated factor of safety.

Figure 12a shows that the potential sliding surface occurred only at the toe position of the slope without seepage, regardless of whether the mean or the 5% fractile or 95% fractile values were taken. The values of the cohesion and friction angle were very different between these three sets of shear parameters, and the effect on the position of the calculated slip shear zone and on the factor of safety was large. The FoS for the 5% strength fractile was less than one, which indicated that the slope would fail at the toe in its current geometry without further triggering, which did not fit the observations. This suggests that the engineering practice of taking a 5% fractile as a cautious estimate for strength parameters is appropriate for engineering purposes but may not be well suited for natural hazard prediction.

For the case of a high groundwater level (GWT3-1458 m), there were some differences in the location of the upper scarp of the landslide obtained with the three sets of strength parameters, but they had little effect on reflecting the overall sliding trend of the upper glacial deposits layer. More importantly, all results were close to the assumed shear zone from the geological investigation. The FoS for the 95% strength fractile was slightly larger than one. This indicated a situation that was very close to the limit state and could be interpreted as highly vulnerable to failure, compared to the FoS of 1.3 normally required by engineers to ensure a long-term stable situation of geotechnical structures.

In general, the results obtained with the mean strength parameters were plausible, and the main mechanical difference between no seepage and high groundwater level was reflected by all three parameter sets. However, a prefailure analysis as a so-called class A prediction [83] would require more samples to reduce the scatter in the determination

of strength parameters. Nonetheless, the following analyses of pore pressure distribution and influence of permeability were conducted with the mean strength parameters as the qualitative influence was expected to be the same for all three parameter sets.

**Table 9.** FoS values for three methods under no-seepage and seepage conditions. Strength parameters were taken as mean values and plausible limits, i.e., 5% and 95% fractiles.

| Initial Groundwater Table | LEA | | | SRFEA | | | FELA [(LB + UB)/2] | | |
|---|---|---|---|---|---|---|---|---|---|
| | Mean | 5% Fractile | 95% Fractile | Mean | 5% Fractile | 95% Fractile | Mean | 5% Fractile | 95% Fractile |
| 0 | 1.30 | 0.90 | 1.54 | 1.37 | 0.90 | 1.63 | 1.38 | 0.95 | 1.66 |
| GWT3-1458 m | 0.93 | 0.73 | 1.09 | 0.94 | 0.71 | 1.07 | 1.01 | 0.78 | 1.16 |

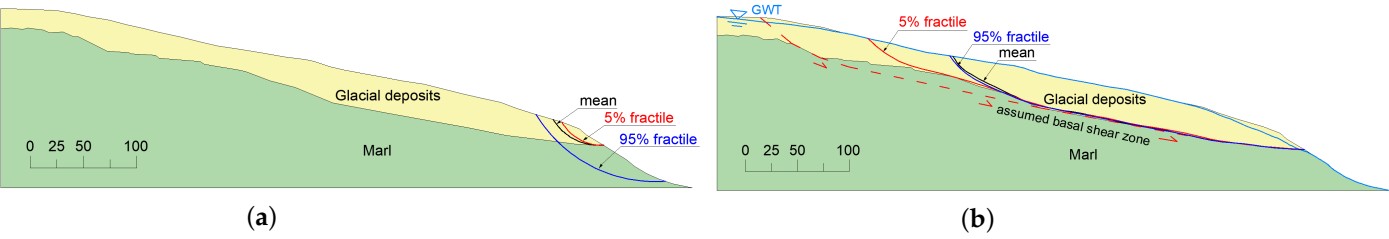

(**a**)                                                 (**b**)

**Figure 12.** Sliding surfaces obtained with mean strength and limit strength (5% and 95% fractiles) by the LEA: (**a**) no-seepage; (**b**) seepage with the initial groundwater table at 1458 m.

The pore pressure distribution at the location of the basal shear zone (marked by the red dashed line in Figure 7a) with different groundwater levels is shown in Figure 13. During the process of the initial groundwater level being raised by tens of meters (the average groundwater level rise in the slope was around 36 m) from GWT1 to GWT3, the maximum value of the pore pressure at the basal shear zone increased from a few tens of kilopascals to around 400 kPa. The increase of the pore pressure reduced the effective stresses and led in combination with the seepage force to the instability of the slope. The different computation methods revealed very similar results in pore water distribution.

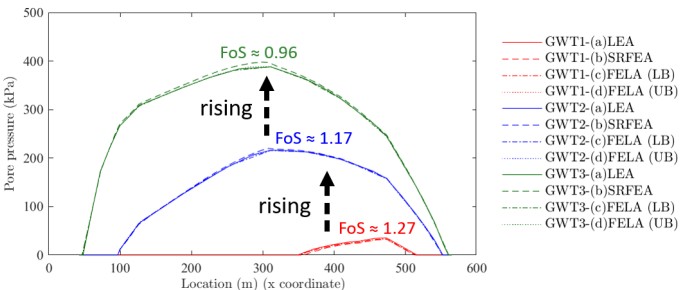

**Figure 13.** The pore pressure distributions with different groundwater levels at the location of the basal shear zone in computations. GWT1, GWT2, and GWT3 represent the initial groundwater tables at 1400 m, 1430 m, and 1458 m, respectively. The origin of the horizontal coordinate represents the left boundary of the calculation model, see Figure 7.

*Effect of Permeability*

The general unfavourable effects of seepage on the slope stability is well known (e.g., [84–86]). However, the special situation of different permeability conditions for different layers requires deeper investigations. To explore the hydromechanical effect on slope stability, we compared our results with a simpler situation, where the coefficients of permeability of the upper and the lower layer were equal (both used $9 \times 10^{-11}$ m/s). Figure 14 shows the resulting sliding surface and the FoS from an LEA computation with the highest groundwater table (GWT3). The FoS of 0.94 was nearly the same as in the

situation with different permeabilities (FoS = 0.93, Table 8); however, the location and shape of the computational shear zone differed largely. Only the lower part near the toe of the slope was affected, whereas the basal shear zone reached further uphill and therefore resembled much more the field-based situation assumed for the computation with different permeabilities.

What was the reason for that? The reason for that lay in the different directions of the groundwater flow in the upper layer. For soils with an isotropic coefficient of permeability, the direction of the flow path was orthogonal to the equipotential lines in Figure 15a, resulting in a more or less parallel flow to the surface. This was quite reasonable for such a long slope which could be approximated as an infinite slope in the middle part. The situation changed when a less permeable upper layer was implemented (Figure 15b). Water flowed through this layer in the direction to the surface, which yielded an upstream in the middle part of the slope. The seepage force also acted in that direction. Then, that force had a component normal to a potential sliding surface and reduced the normal stress and therefore the frictional resistance. The same effect could be seen by observing the pore water distribution along the geologically assumed shear zone, as shown in Figures 16 and 17. The pore water pressure in the upper part of the slope was considerably higher for the computation with different permeabilities than for the computation with equal permeabilities. This led to a stronger reduction of the effective normal stresses and in turn to a stronger reduction of the shear strength, which enabled the sliding surface to expand more uphill.

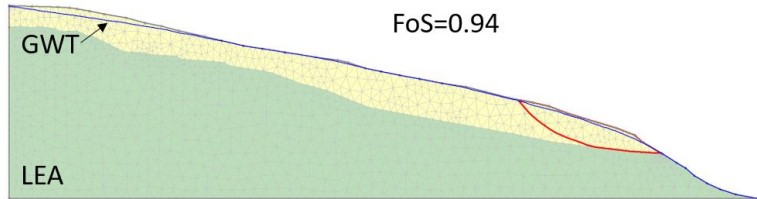

**Figure 14.** Sliding surface obtained with the same permeability coefficient for both layers by the LEA (GWT3).

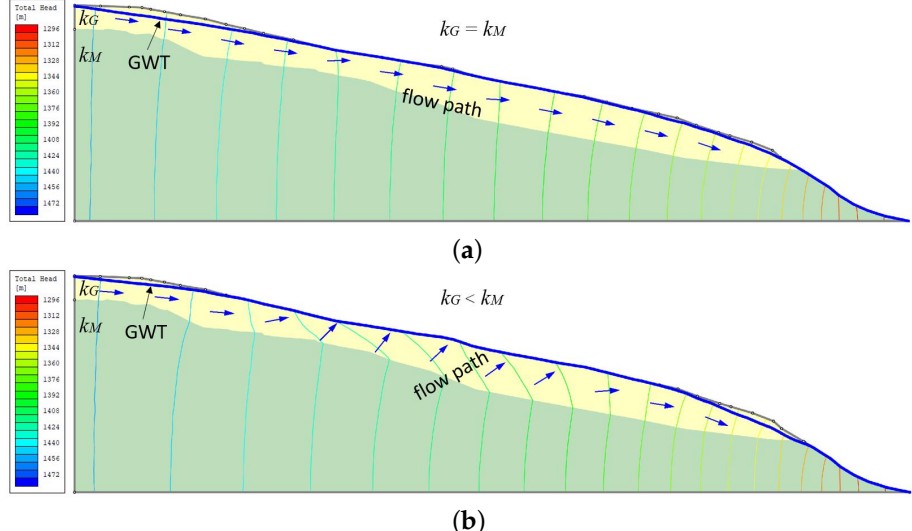

**Figure 15.** Hydraulic head distributions and directions of the groundwater flow (in the upper glacial deposit layer). $k_G$ and $k_M$ represent the permeability coefficients in the upper glacial deposit and lower marl layers, respectively: (**a**) in the case of equal permeability ($k_G = k_M$); (**b**) in the case of different permeabilities ($k_G < k_M$).

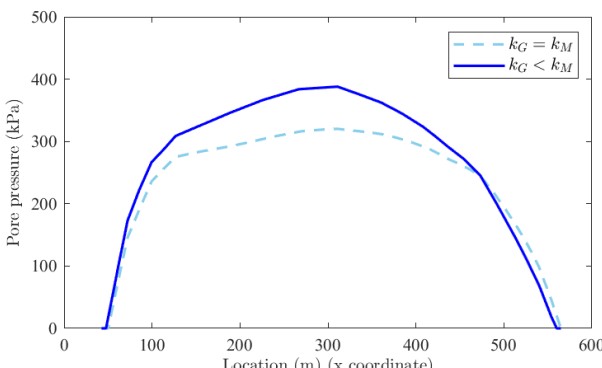

**Figure 16.** The pore pressure distributions obtained with different permeability coefficient settings for upper and lower layers at the location of the basal shear zone by the LEA.

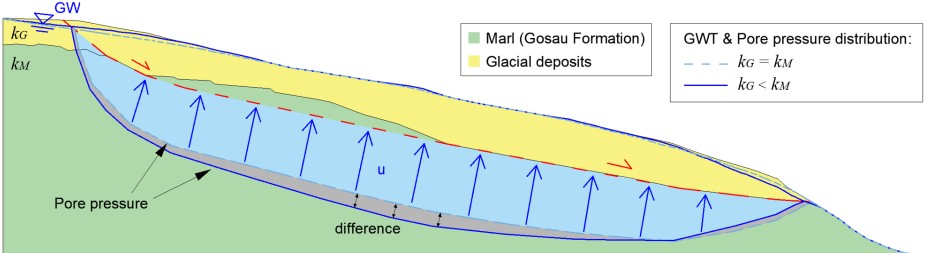

**Figure 17.** Schematic diagram of pore pressure distributions at the location of the basal shear zone (red dotted line) for cases with the same and different permeabilities for both layers.

## 6. Discussion

### 6.1. Unique Geological Situation

The occurrence of the Ludoialm landslide increases our concern on the stability of relatively flat slopes. Deep-seated landslides in Alpine mountainous regions are usually observed on slopes with a dip angle of the slope greater than 20°, e.g., the Gradenbach landslide, the Reissenschuh landslide, or the Séchilienne landslide. The sliding mechanisms of these deep-seated landslides [9–13] differ significantly from the case investigated in this contribution. A case that also occurred in the Alps, the Gschliefgraben mudslide [87,88], is somehow comparable to the Ludoialm landslide. The Gschliefgraben mudslide also occurred on a gentle slope with a dip angle less than 10°, with alternating layers of less permeable and more permeable soils. This event was presumably triggered by a rockslide in the catchment area together with humid weather [88] and differs considerably from the case in this contribution, which was most likely caused by the pore pressure rise generated by snow melting.

The layer of glacial sediments that covers the Ludoialm landslide area has a considerable thickness, which indicates that the soil is very weak from the slope surface down to a relatively large depth. A large thickness of glacial sediments that represent a weak strength is rarely observed for the geological compositions in most alpine deep-seated landslide areas [9–11], which could also be one contributing factor for the reactivation of the Ludoialm landslide. In this case, the unique features of this landslide can be summarized in the following two aspects.

Firstly, the difference of permeabilities between the upper glacial deposits layer and the lower marl layer has a great influence on the stability of the flat slope, whose hydromechanical effect can be seen in Figures 15 and 17.

Secondly, the shear strength of glacial deposits in the uppermost layer of the Ludoialm landslide is relatively low. The friction angle of about 19° is smaller than that of the glacial till in most cases, which is around 30°, or at least larger than 20° [89–93]. This weak material fosters the instability of the slope, which was reactivated twice in the past decades even

though the slope is quite gentle with the dip angle smaller than the soil friction angle of the sliding mass.

The overall shear strength at the slope scale can also be estimated by a backward analysis of a failure. This could be more reliable than using a limited number of laboratory tests in some cases. However, in this case, we were focusing on the mechanical cause of the failure, and since the shear strength was only one of the parameters affecting the failure, a backward analysis would probably hide the cause of the failure and its mechanism.

*6.2. Shear Zone*

Due to the lack of drillings, it is challenging to observe the basal shear zone of a landslide with geological investigation, as this zone is only visible at scarps and the lateral flanks. Computational methods may help understand the shear zone evolution and the geometry of the shear zone. The pore pressure is an important indicator to reflect the process of shear band evolution [94]. In this case study, the variation of pore pressure in the basal shear zone due to the change of groundwater level was clearly reflected in the calculation. Compared to the lower marl layer, a lower permeability of the upper glacial deposits layer was verified by calculation that could lead to significant changes of groundwater flow compared to a situation with the same permeability in both layers. The seepage force in the shear zone acted in the direction of the water flow (Figure 15b) and was approximately perpendicular to the zone pointing upwards to the slope surface. This is similar to a lifting force for the upper soil, enabling landslides to occur more easily.

To sum up, the unique geological and permeability conditions of the Ludoialm landslide area, combined with the rising of groundwater due to the rapid snow melting, may have led to the initial formation of the landslide and following reactivations. Figure 18 shows the schematic diagram of the Ludoialm landslide failure mechanism, especially the evolution of the shear band. The increase of the seepage force at the basal shear zone leads to a decrease of shear resistance, which is the key point of the slope collapse.

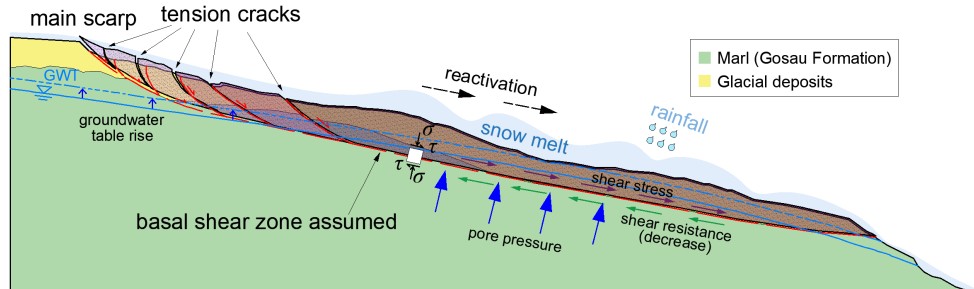

**Figure 18.** Schematic diagram of the landslide failure mechanism (size of deformation is exaggerated due to readability reasons including tension cracks).

*6.3. Mitigation Measures*

Since there are almost no settlements or major roads in the surroundings of this landslide, except a farm above, the risk of reactivation is not high. However, the soil of the landslide mass at the toe can be transported down to the bottom of the valley over a long distance, which is a debris flow when the water content in the soil is high.

A monitoring and early warning system, including but not limited to regular site visit or geological survey, is an effective measure for the detection of critical phases or accelerations [95]. Moreover, some engineering approaches can be considered, e.g., drainage. Actually, several drainage trenches have been built since the reactivation of the landslide in 1967. After that, some additional draining pipes were installed to increase the system functionality, although some of them have been destroyed by the movement of the landslide.

## 7. Conclusions

Geological and geotechnical investigations on the Ludoialm landslide were provided, to show the geology, the geomorphology of this slope, the changes in the slope geometry before and after the landslide event, and the characteristics of each soil layer. Computations with three methods (LEA, SRFEM, and FELA) simulated the process of the initial slope failure and reproduced well the basal sliding zone obtained from the geological investigations.

The calculation results verified that the initial failure of the slope was most likely caused by the rise of the groundwater level generated by snow melting, possibly intensified by rain. The raising groundwater level increased the pore water pressure at the basal sliding zone, and the difference of permeabilities between glacial deposits and marl changed the groundwater flow direction and thus the seepage force in an unfavourable direction. The slope lost its stability due to the increased pore pressure when the groundwater level approached the slope surface, triggering this large-scale landslide. In addition, considering different coefficients of permeability was essential to achieve a realistic geometry of the basal shear zone. The scatter in strength was quite large due to the limited number of specimens available. This did not affect the overall conclusion about the possible cause of failure in this postfailure analysis, but the safety factors could not be calculated with high confidence. Therefore, a much larger number of specimens would be required in a prefailure analysis.

The occurrence of the Ludoialm landslide is highly related to its own unique geological situation, in addition to external triggering factors. The present paper provided a valuable reference for studying the shear band evolution and failure mechanism of landslides in exceptionally low inclined large-scale slope.

**Author Contributions:** Conceptualization, X.D., B.S.-M., W.F. and C.Z.; methodology, X.D., B.S.-M. and W.F.; software, X.D.; validation, X.D.; formal analysis, X.D., B.S.-M. and W.F.; investigation, X.D.; resources, B.S.-M., J.K. and C.Z.; data curation, X.D.; writing—original draft preparation, X.D. with partial contribution of C.Z. and W.F.; writing—review and editing, X.D., B.S.-M., J.K., C.Z. and W.F.; visualization, X.D.; supervision, B.S.-M. and W.F.; project administration, B.S.-M. and W.F.; funding acquisition, X.D., B.S.-M. and W.F. All authors have read and agreed to the published version of the manuscript.

**Funding:** This research was funded by the University of Innsbruck in support of the doctoral program "Natural Hazards in Mountain Regions". Xiaoru Dai has also received funding from the University of Innsbruck's "Exzellenzstipendien für Doktoratskollegs" fellowship programme (Grant No. 2021/TECH-47) and the Tyrolean Science Fund Projects (Project No. F.45084). Open access funding is provided by the Vice Rectorate for Research of the University of Innsbruck.

**Institutional Review Board Statement:** Not applicable.

**Informed Consent Statement:** Not applicable.

**Data Availability Statement:** The data that support the findings of this study are available from the corresponding author upon reasonable request.

**Acknowledgments:** We thank the three anonymous referees for their constructive comments, which helped to improve the presentation of this paper.

**Conflicts of Interest:** The authors declare no conflict of interest.

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
