# Peer review of "Mechanisms for the Formation of an Exceptionally Gently Inclined Basal Shear Zone of a Landslide in Glacial Sediments—The Ludoialm Case Study"

_applsci, doi:10.3390/app13116837_

Round 1
Reviewer 1 Report
This article presents a numerical investigation and analyses of possible mechanism for a gently inclined landslide. The topic might be interesting. Unfortunately, the manuscript presented did not show scientific significance to the readers interested. I outline my concerns below.
(1) Is the simulation results verified by the field observed data?
(2) What is the purpose of using three different methods for slope stability analyses here? I don’t think they are able to reveal the failure mechanism of this landslide due to considerably simplified assumptions during calculation.
(3) The mechanical behaviors of the shear zone are important, but missing information in the text.
(4) The discussion is too general an does not reveal the mechanism behind the landslide.
(5) Conclusion: The applicability of the mentioned two methods is well established, and certainly valid for investigation of large-scale landslides. It should not be listed as the conclusion of this study.
Reviewer 2 Report
This manuscript studies the failure mechanisms of a landslide in glacial sediments using LEA, SRFEM, and FELA methods. The paper was generally well organized. The results of the study could be interesting for landslide hazards. I do recommend the manuscript to publish in Applied Sciences after major revision. There are some minor revisions according to the following comments:
1) More significant contributions need to be added to the manuscript and research to be accepted as a journal paper in the revised version. For example, the abstract needs to write qualitatively and mention the results of the paper. Please explain more about the results and benefits of the manuscript in the abstract and also the conclusions.
2) Please provide details of numerical analysis including mesh size, number of mesh, and mesh sensitivity analysis.
3) Please present the water flow by vector and discuss the effect of water flow on the shear zone formation (Figs. 11 and 12).
4) Was the phreatic level used for pore water pressure calculation or steady-state groundwater flow?
5) More interpretation of the results needed to be added to the manuscript.
6) It is recommended to perform a sensitivity analysis for the permeability coefficient and water table.
7) It is also recommended to include the mitigation technique of landslide in glacial sediments
Reviewer 3 Report
The submitted paper still lacks of a profound internal review, as typing and comma errors were submitted. The enclosed comments are not exhaustive.
Generally, both case study and methodology are of high interest.
The only criticism is about field sampling and bias. I think, this study is really excellent and a longly desired numerical prove of widely observed phonemona .... not only in Gschliefgraben, but even in Southamerican Andeas or Asian Tien Shan mountains. Therefore, maybe the number of samples and their laboratory results is underpresenting this important phenomena and inadvertendly decreasing the value of this - generally - highly valued study. Please consider this in your revision: Do not base this so important and so desired paper on five sketchy laboratory analysis! Please do not take the average of three cohesion values between 0 and 59 - these points cannot be accepted, but the rest of this paper is very important and shall be published as soon as possible.

Round 2
Reviewer 2 Report
Thanks for revising the manuscript. The current form of the manuscript could be published.
Author Response
Thanks for your comments!